# Vertical bearing performance of inclined high-pressure rotary spray pile

Shilang Guo[1], Fei Gan [1]*, Hong Wang[1], Jing Bi[2], Biao Liu[3], Yuanyin Zhang[3]

**1** School of Civil Engineering, Guizhou University, Guiyang, China, **2** Guizhou Key Laboratory of Geotechnical Mechanics and Engineering Safety, Guiyang, China, **3** Guizhou Power Transmission and Transformation Limited Liability Company, Guiyang, China

* fgan@gzu.edu.cn

## Abstract

Inclined piles are increasingly adopted in the foundation design of large-scale transmission line towers. Nine inclined pile tests were conducted in red clay soil to investigate the load-bearing and deformation behavior and the load transfer mechanism of inclined piles under vertical loading. The effects of pile inclination and length-to-diameter ratio on axial force, bending moment, shear force, and lateral friction force were analyzed. The test results indicate that: (1) As the inclination angle increases, the settlement of the inclined pile increases under vertical loading. (2) The axial force in the inclined pile is smaller compared to that of the corresponding vertical pile with the same material and dimensions. (3) Bending moment and shear force are observed in the upper section of the inclined pile, with the maximum bending moment influenced by both the pile's inclination angle and length-to-diameter ratio. (4) In the upper section of the pile, the average soil-side frictional resistance of inclined piles is higher than that of vertical piles. The maximum resistance occurs in the section from the top of the pile to a relative depth (Z/L) of approximately 0.14.

## 1 Introduction

The rapid development of China's power industry has driven the need for transmission line construction as the current transmission system is gradually becoming congested. However, the potential hazards associated with perennial strong winds and snowstorms affect the safety and stability of these transmission lines [1,2]. Therefore, Electric tower foundations must endure large compressive loads (Fig 1). Due to the uncertainty in the wind load direction, transmission tower foundations are subjected to cyclic compressive loads, with compressive capacity being the critical factor for ensuring foundation stability. Additionally, it is essential to consider the economic viability of tower base construction. The magnitude and frequency of loads on tower foundations may increase as a result of extreme storm events driven by climate change, raising significant concerns about the long-term performance of the foundation system. Various types of structures are currently used for the construction of transmission tower foundations, such as piles [3–5], anchors [6–8], inverted T-shape strip footing [9,10], pier foundations [11,12], and helical piles [13–15], etc. High-pressure rotary piles provide high bearing capacity, are easy to construct, and cause less damage to the ecological environment. Therefore, they are widely used in transmission line projects in the mountainous areas

**Data availability statement:** All relevant data are within the manuscript and its Supporting Information files.

**Funding:** This work was supported by the National Natural Science Foundation of China (Grant No. 52164001). The funders had no role in study design, data collection and analysis, decision to publish, or preparation of the manuscript.

**Competing interests:** The authors have declared that no competing interests exist.

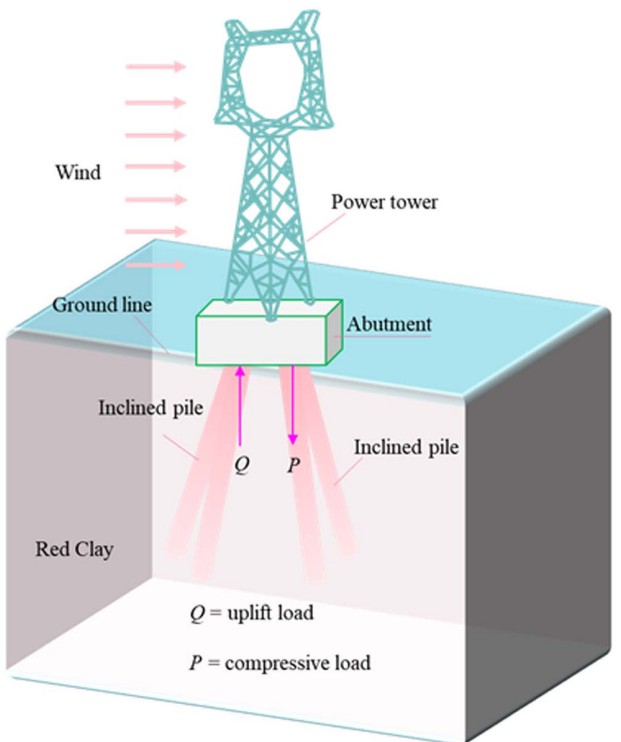

**Fig 1. Power tower pile foundation.**

of western China. In the past decades, many studies have been conducted to investigate the bearing behavior of high-pressure rotary piles [16–18]; for example, Shibata [19] and Hanna et al. [20] conducted a preliminary study of lateral friction resistance of inclined monopiles in homogeneous sandy and homogeneous soft soils by modeling tests. Still, they did not consider that the friction resistance exerts its effect along the depth and regarded the friction resistance as the same throughout the entire length of the pile, which is not by the actual situation. Shen [21] and Wang et al. [22] derived the formula for calculating the pile diameter of rotary piling construction parameters and soil parameters controlled by the single-pipe method. They summarized the formula for calculating the bearing capacity of the al. single pile. Li and Yun et al. [23, 24] analyzed the pile lateral frictional res, instance, pile bending moment, and pile shear force of inclined piles by numerical simulation and theoretical analysis, respectively. It is worth noting that most of these studies were conducted to investigate the behavior of compressive piles using sand (with varying dense compaction) or cohesionless soils. However, they did not consider spatial soil type mapping and the corresponding effects on the results [25] and did not assess the impact of clay sensitivity on engineering results [26–29]; many transmission tower foundations are embedded in clay. However, insufficient research engineering design experience on the bearing characteristics of compressive piles in red clay remains to be improved.

The above analysis indicates that current experimental research on high-pressure rotary piles primarily focuses on their bearing capacity [30–34]. However, fewer studies have been conducted on the load transfer mechanism and internal forces in high-pressure rotary spray piles. Additionally, the effects of the length-to-diameter ratio and inclination angle on the working properties of inclined piles remain underexplored. Studies on factors such as axial force, lateral frictional resistance, bending moments, and shear forces in inclined piles are

mainly conducted through numerical simulations and theoretical analysis, with a noticeable lack of necessary experimental research. This paper presents a model test of an inclined pile in a red clay foundation under vertical load. It systematically analyzes the impact of the pile inclination angle and length-to-diameter ratio on the axial force, bending moment, shear force, and average frictional force along the pile side. The findings of this study are of significant practical importance for advancing design theories and guiding engineering construction.

## 2  Model tests

### 2.1  Test equipment

The model tank and the associated test apparatus are shown in Fig 2. The model tank has dimensions of 1.2 m × 1.2 m × 1.5 m (length × width × height) and is primarily constructed from a steel frame and tempered glass with a thickness of 10 mm. The fill material was compacted in layers using artificial compaction methods. Data acquisition was performed using a static strain collector and a PC. The loading was applied by introducing a reaction force through a jack (YCW30A-80 type) and a reaction beam. Additionally, two displacement gauges were symmetrically positioned on the bearing platform to monitor vertical displacement. A schematic diagram of the overall test setup is provided in Fig 2.

The data acquisition devices are;

(1)  Electronic Percentage Meter. After placing the pile inside the model box and pouring the bearing platform, an electronic percentage meter was fixed on the bearing platform to monitor the pile top displacement, as shown in Fig 2.

(2)  Strain gauges. Strain gauges were symmetrically placed on each pile section to measure the bending moment and axial force during the test, as shown in Fig 3.

(3)  Jack hydraulic gauge. As shown in Fig 2, the reading was allowed to stabilize at the end of each loading cycle before recording the value.

(4)  The electronic static strain gauge uT7130 was used, with the following specifications: maximum range ± 30,000 $\mu\varepsilon$; measurement accuracy: 0.1 $\mu\varepsilon$; and measurement error: ± 1‰.

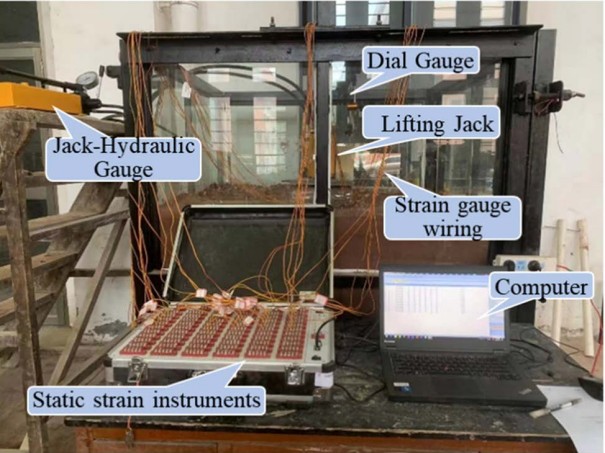

**Fig 2. Diagram of the test loading device.**

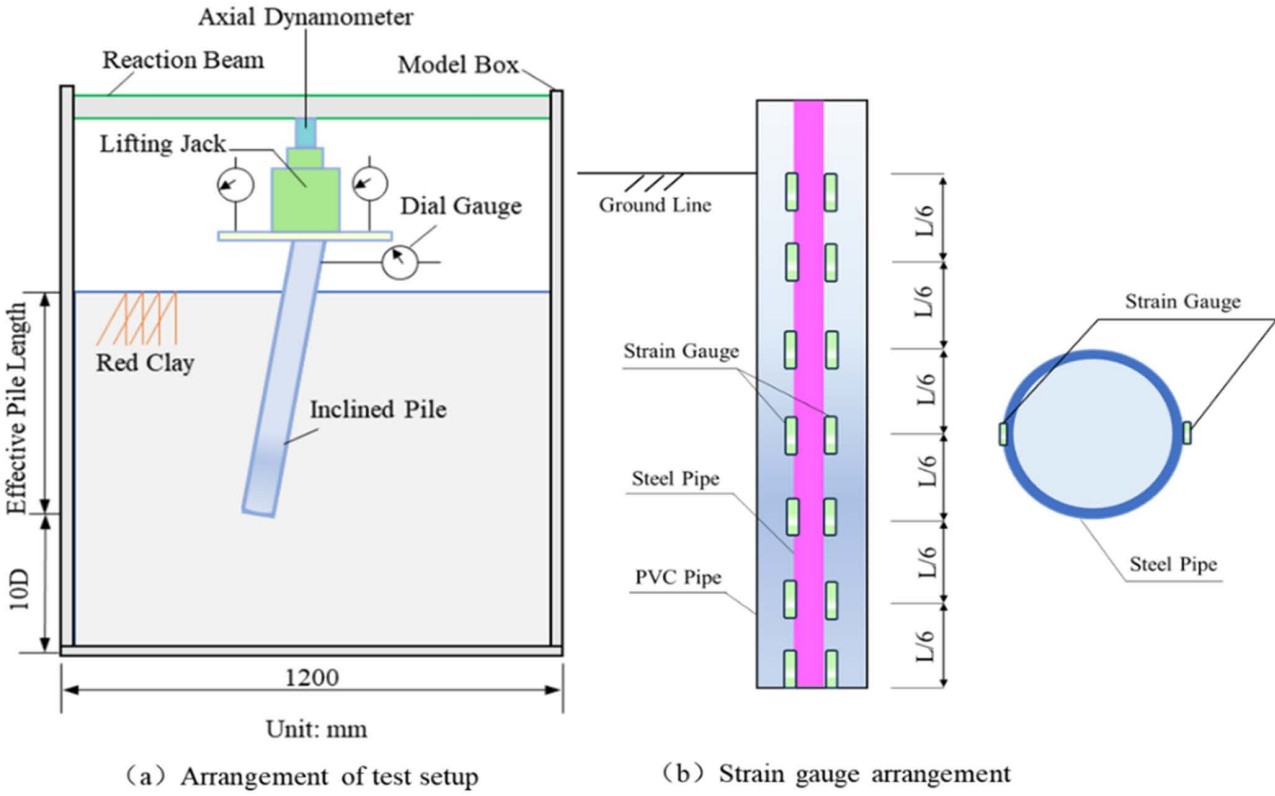

**Fig 3. Model tests.**

## 2.2 Test program

A total of 9 experiments, organized into three groups, were designed in this study, with the experimental protocol shown in Table 1. Three inclined piles with different inclination angles were tested in each group to evaluate the effect of pile inclination on the pullout resistance. All piles had the same length-to-diameter ratio, which allowed for the investigation of the influence of pile inclination on pullout resistance. In each of the three test sets, three inclined piles with the same inclination angle were tested to directly compare the effect of varying length-to-diameter ratios on the compressive properties of the inclined piles. The schematic of the model test arrangement is shown in Fig 3.

**Table 1. Test program.**

| Grouping | Number | Angles/° | L/mm | D/mm | L/D ratio |
|---|---|---|---|---|---|
| T1 | T11 | 0 | 660 | 40 | 16.5 |
| | T12 | 10 | 660 | 40 | 16.5 |
| | T13 | 20 | 660 | 40 | 16.5 |
| T2 | T21 | 0 | 860 | 40 | 21.5 |
| | T22 | 10 | 860 | 40 | 21.5 |
| | T23 | 20 | 860 | 40 | 21.5 |
| T3 | T31 | 0 | 1060 | 40 | 26.5 |
| | T32 | 10 | 1060 | 40 | 26.5 |
| | T33 | 20 | 1060 | 40 | 26.5 |

## 2.3 Test materials

The model piles were fabricated using cast-in-place slurry reinforcement techniques in hydraulic soil, with the outer diameter of the piles being 40 mm. Fabricating the model pile begins by burying the PVC pipe at the predetermined location and then applying a smooth plastic film around the PVC pipe to facilitate the later demolding process. After the filling was completed, the PVC pipe was withdrawn, and a prefabricated thin-walled hollow aluminum pipe (outer diameter: 15 mm, wall thickness: 2 mm) was inserted into the pile hole to simulate the reinforcing steel. To ensure accurate strain measurements, strain gauges were symmetrically attached to both sides of the alloy tube. During the placement of the alloy tube, a 35 mm diameter steel ring was used at the end of the pile to ensure the tube was positioned centrally within the hole. Strain gauges and connecting lines were applied at 100 mm intervals along the inner wall of each tube, from the top to the bottom of the pile. The connecting lines were securely bonded with 502 glue, and key nodes were coated with epoxy resin for protection. Two strain gauges were symmetrically attached at the exact cross-sectional location of the model pile, and the average of their readings was used as the strain value for the pile at that cross-section. Refer to Fig 3 for the arrangement of the strain gauges on the model pile.

The soil used in this test was taken from the red clay soil of a slope in Guizhou University, and the amount of pre-filled soil for this test was 2 m³. Before filling the soil for the test, the soil was sampled. According to the Standard for Geotechnical Test Methods (GB/T 50123-2019) [35], the soil samples' fundamental physical indicators measured the soil's relevant physical parameters, as shown in Table 2. The particle gradation curve of the test soil was obtained by gradation sieving, as shown in Fig 4.

**Table 2. Indicators of basic physical and mechanical properties of test soil samples.**

| $\rho$/(g/cm³) | $\rho$dmax/cm³) | $w$/% | $W_L$/% | WP/% | Cu | Cc |
|---|---|---|---|---|---|---|
| 1.82 | 1.69 | 35.2 | 59.47 | 25.72 | 6.09 | 0.4 |

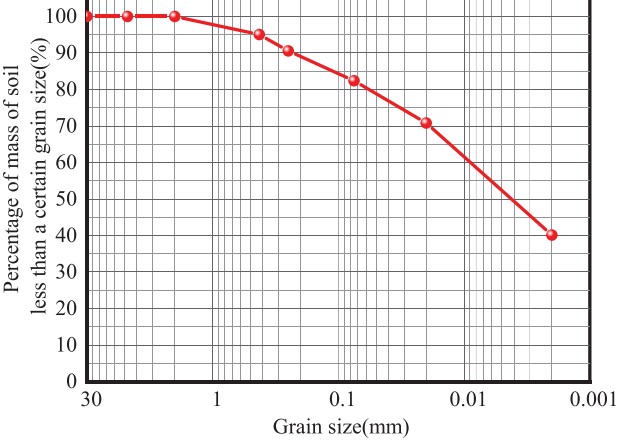

**Fig 4. Grain size distribution of red clay.**

## 2.4  Test steps

The holding layer at the bottom of the pile and the soil on the side of the pile were made of red clay. The model tank was filled in layers using the sand-and-rain method, with each layer having a thickness of 200 mm. After each layer was filled, the soil with a density of 1800 kg/m³ was compacted using a vibratory smoothing machine. First, fill the clay holding layer at the bottom of the model tank to a thickness of 10D (where D is the diameter of the model pile). After completing the holding layer, fix the model pile. After placing the pile, wrap a smooth plastic film around the side to reduce lateral friction when the pipe is pulled out later. After the filling is complete, slowly rotate and remove the pre-embedded PVC pipe. Then, insert a thin-walled hollow aluminum pipe with affixed strain gauges into the hole. This pipe will serve as the force reinforcement for the rotary spray pile. A protective 45 mm diameter rebar is welded to the end of the alloy pipe to ensure the pipe remains centered in the hole during the soil pouring process. Use well-mixed soil for pouring, providing that acceptable, dense soil is consistently used during the process. The soil filling coefficient should be no less than 1.05. The clay piles were cast and left in the model tanks for 7 days to reach the design strength under conditions similar to those in the in-situ tests.

After setting up the test device, the inclined pile will be tested once the settlement of the pile and surrounding soil under self-weight is less than 0.01 mm per 24 hours. By the code [36] and relevant literature on model test stability criteria [37–39], the slow load maintenance method was used to determine the vertical compressive ultimate bearing capacity (Q) of the single pile. During loading, each load increment is set to 1/10 of the estimated ultimate load. If the settlement and horizontal displacement of the pile top are both less than 0.01 mm within 10 minutes, and the load maintenance time is at least 30 minutes, the next load increment can be applied. Load stabilization is considered when two consecutive load increments meet these criteria, and the process continues until the settlement reaches the failure threshold, at which point loading is stopped. The load-displacement curves for the model piles obtained during the tests are shown in Fig 5

When the pile top load reaches 8 kN (Level 8), the pile stabilizes, but when the load is increased to 9 kN (Level 9), a significant amount of settlement occurs at the top of the pile. This settlement is 4.35 times greater than that observed at the 8 kN load level, indicating that the pile is approaching failure or damage. Consequently, the design load (Q) is taken as 8 kN,

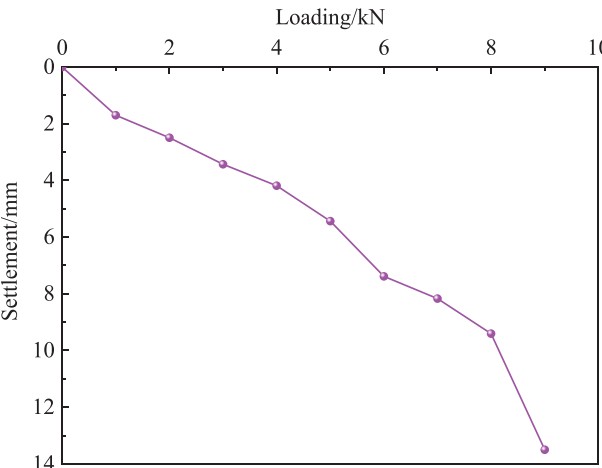

**Fig 5.  Load-displacement relationship curve.**

with a safety factor 2 applied to ensure stability. Therefore, the working load for a single pile is determined to be 4 kN, which is half of the maximum load to provide an adequate safety margin.

## 3 Analysis of test results

### 3.1 Horizontal bearing capacity

As can be seen from Fig 6, no lateral displacement occurred in the vertical pile under vertical load. The horizontal displacement of the top of the inclined pile under the same vertical load increases with the increase of vertical load, and the increase of the horizontal displacement of the top of the pile is smooth; for example, the horizontal displacements of the pile tops of the T22 group under 1 ~ 4 levels of vertical loads were 1.280 mm, 3.381 mm, 5.089 mm, and 7.045 mm, with increases of 164.14%, 50.52%, and 38.44%, respectively, compared to the previous load level. For the T32 group, the displacements were 3.800 mm, 5.881 mm, 5.061 mm, and 8.000 mm, with increases of 54.76%, 20.07%, and 13.30%, respectively, compared to the previous load level. Under the load stabilization condition, the displacement of the pile top increases with the inclination angle, indicating a decrease in the vertical bearing performance of the pile foundation. For example, the displacements for groups T13 and T12 were 6.802 mm and 6.725 mm, respectively. For groups T23 and T22, the displacements were 7.522 mm and 7.045 mm, respectively, and for groups T33 and T32, the displacements were 8.45 mm and 8.00 mm, respectively. Under the same conditions, the larger the inclination angle, the greater the horizontal displacement of the inclined pile. This occurs because the vertical load generates a horizontal component, disturbing the soil structure on the side of the pile. This disturbance reduces or even eliminates the soil's reaction force, which ultimately diminishes the pile foundation's bearing performance.

The larger the length-to-diameter ratio of the inclined pile, the more pronounced the change of horizontal displacement of the pile top, such as the horizontal displacements of the pile tops for the T12, T22, and T32 groups at steady-state settlement were 6.725 mm, 7.045 mm, and 8.000 mm, respectively, compared to the T12 group, the horizontal displacement at the pile top for the T22 and T32 groups increased by 4.76% and 13.56%, respectively. Under the same length-to-diameter ratio, a larger tilt angle of the rotary spray pile results in a

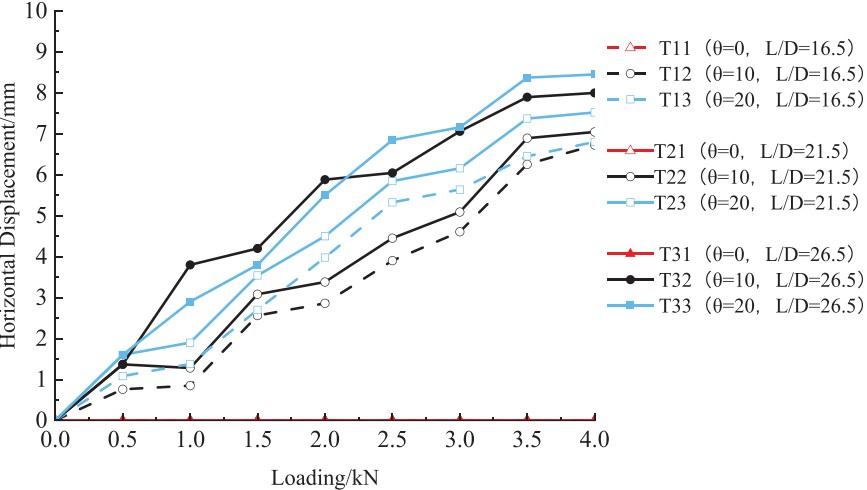

**Fig 6. Change curve of horizontal displacement of pile top.**

more pronounced rate of change in the horizontal displacement at the pile top. For example, when L/D = 26.5 and the pile top load increases sequentially from 1 kN to 4 kN, the horizontal displacements at the pile top for the T33 group were 2.900 mm, 5.500 mm, 7.159 mm, and 8.450 mm, representing increases of 89.66%, 30.16%, and 18.03%, respectively, compared to the previous load level. When comparing the T33 group with the T32 group under the same vertical load and length-to-diameter ratio, it is evident that the rate of horizontal displacement changes increases with the inclination angle. The greater the tilt angle, the larger the rate of horizontal displacement change.

## 3.2  Vertical bearing capacity

As shown in Fig 7, the settlement displacements of inclined piles increase with vertical load. As the load on the pile top increases, the settlement difference between inclined and vertical piles also grows, which finally leads to the settlement of the top of inclined rotary spray piles being more significant than that of vertical piles under the same conditions. For example, at a vertical load of Q = 1.5 kN, the pile top settlements for the T21, T22, and T23 groups were 1.141 mm, 1.047 mm, and 0.998 mm, respectively. The settlements of T22 and T23 were 14.32% and 4.91% greater than that of T21. At Q = 4 kN, the pile top settlements for T21, T22, and T23 groups were 6.086 mm, 5.722 mm, and 5.091 mm, respectively. The settlement of the T22 and T23 groups is 19.54% and 12.39%, respectively, compared with that of the T21 group. This demonstrates that as the load on the pile top increases, the settlement difference between inclined and vertical piles also becomes more pronounced under the same test conditions.

For a given length-to-diameter ratio, increasing the inclination angle of the pile reduces the vertical bearing performance of the pile foundation. For example, when L/D = 16.5, the pile top settlements for the T11, T12, and T13 groups at steady state were 6.062 mm, 6.722 mm, and 7.232 mm, respectively. The settlements of the T12 and T13 groups were 10.89% and 19.30% higher than those of the T11 group, respectively. This indicates that, under the same test conditions, a larger inclination angle results in more significant vertical displacement at the pile top. This suggests that reducing the inclination angle to a certain extent can enhance the vertical bearing performance. Specifically, the vertical compressive ultimate bearing capacity of inclined piles is lower than that of vertical piles [40].

A comparative analysis of the change curves of pile top settlement displacement with the length-diameter ratio in Fig 7(a)–(c) shows that, for the same vertical load on the pile top, a more considerable length-diameter ratio results in a smaller pile settlement. When the vertical load on the pile top is stabilized at 4 kN, the pile top settlements for the T13, T23, and T33 groups are 7.232 mm, 6.086 mm, and 5.486 mm, respectively. The settlement of T33 is 75% and 90% of the settlements of T23 and T13, respectively, indicating that the vertical bearing performance of the pile with L/D = 26.5 is superior to that of the piles with L/D = 16.5 under the same test conditions. Thus, under the same test conditions, the bearing performance of the pile with L/D = 26.5 (T33) is superior to that of the piles with L/D = 16.5 (T13 and T23).

For a constant vertical load, the settlement of the pile top decreases as the pile length-to-diameter ratio increases, indicating that a higher length-to-diameter ratio can enhance the vertical bearing performance of inclined rotary spray piles under certain conditions. The vertical bearing capacity of an inclined rotary spray pile is influenced by the horizontal component of the vertical load and the soil reaction force on the pile's side. As the vertical load acts at an angle, it generates a horizontal force, which can disrupt the soil structure along the pile side, leading to a reduction or even loss of the soil reaction force. For a constant length-to-diameter ratio, the pile top settlement increases with the inclination angle, suggesting that a higher inclination angle reduces the vertical bearing performance

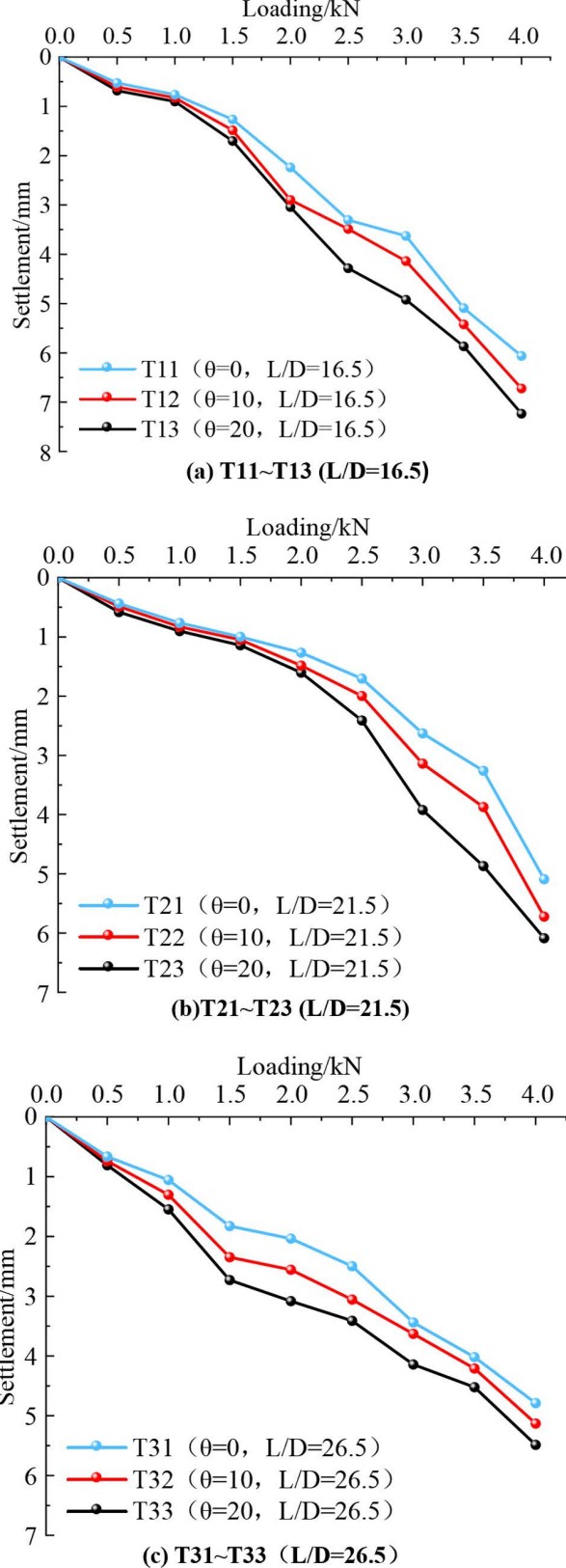

**Fig 7. Variation curve of pile top settlement displacement.**

of the pile foundation. As the inclination angle increases, the vertical load generates a more significant horizontal component force. This reduces the vertical force acting on the pile and subjects it to more excellent bending and shear forces. The more significant horizontal force increases the contact between the pile and the soil on its side, enhancing frictional resistance and ultimately reducing the vertical settlement.

Table 3 shows that after the pile top settlement stabilizes under a 4 kN mechanical load, the horizontal bearing capacity of inclined piles is always smaller than that of vertical piles. Additionally, the vertical bearing capacity decreases as the inclination angle of the pile increases. For instance, with an L/D ratio of 21.5, the rate of change for T23 and T22 compared to T21 is 12.39% and 19.54%, respectively. When the L/D ratio is 26.5, the rate of change for T33 and T32 compared to T31 is 7.76% and 7.10%, respectively. Under the same test conditions and pile inclination angle, as the L/D ratio increases, the rate of change in pile settlement decreases. This indicates that increasing the L/D ratio of the rotary spray pile enhances its vertical bearing capacity. The increase in the L/D ratio enlarges the contact surface between the pile and the surrounding soil, leading to a more significant change in frictional resistance along the pile side.

## 3.3 Axial forces

The axial force of the pile can be calculated according to formula (1) [41]:

$$N_i = EA\varepsilon_i \tag{1}$$

where $E$ is the modulus of elasticity, $A$ is the cross-sectional area of the pile, $\varepsilon_i$ is the strain value at measurement point $i$. It is stipulated that the axial force value of the pile is positive when under pressure.

Fig 8 shows that the axial force of the pile decreases with increasing adequate depth under different working conditions. The rate of change is small under smaller mechanical loads (Q < 2 kN) but becomes more pronounced with larger mechanical loads (Q > 2 kN). For instance, when the top of the T11 pile is subjected to a 1 kN load, the axial force is 0.72 kN at Z/L = 0.14 and 0.11 kN at Z/L = 0.85, a reduction of 0.61 kN. Under a 4 kN load, the axial force is 3.38 kN at Z/L = 0.14 and 0.829 kN at Z/L = 0.85, a reduction of 2.551 kN. Under smaller mechanical loads, the pile side friction force is more effective, resulting in minimal relative displacement between the pile and soil, and the pile end force is less significant. This leads to a more minor overall reduction in axial force. As the mechanical load increases, the relative displacement between the pile and soil increases, causing the pile end force to grow. Consequently, the axial force change becomes more gradual and significant.

**Table 3. Settlement change rate of the inclined pile with different loading conditions.**

| Grouping | Number | Loading/kN | Settlement/mm | Rate of change of settlement | L/D ratio sedimentation rate of change |
|---|---|---|---|---|---|
| T1 | T11 | 4 | 6.062 | / | / |
| | T12 | 4 | 6.722 | 10.89% | / |
| | T13 | 4 | 7.232 | 19.30% | / |
| T2 | T21 | 4 | 5.091 | / | 19.07% |
| | T22 | 4 | 5.722 | 12.39% | 17.47% |
| | T23 | 4 | 6.086 | 19.54% | 18.83% |
| T3 | T31 | 4 | 4.79 | / | 26.56% |
| | T32 | 4 | 5.13 | 7.10% | 31.03% |
| | T33 | 4 | 5.486 | 7.76% | 31.83% |

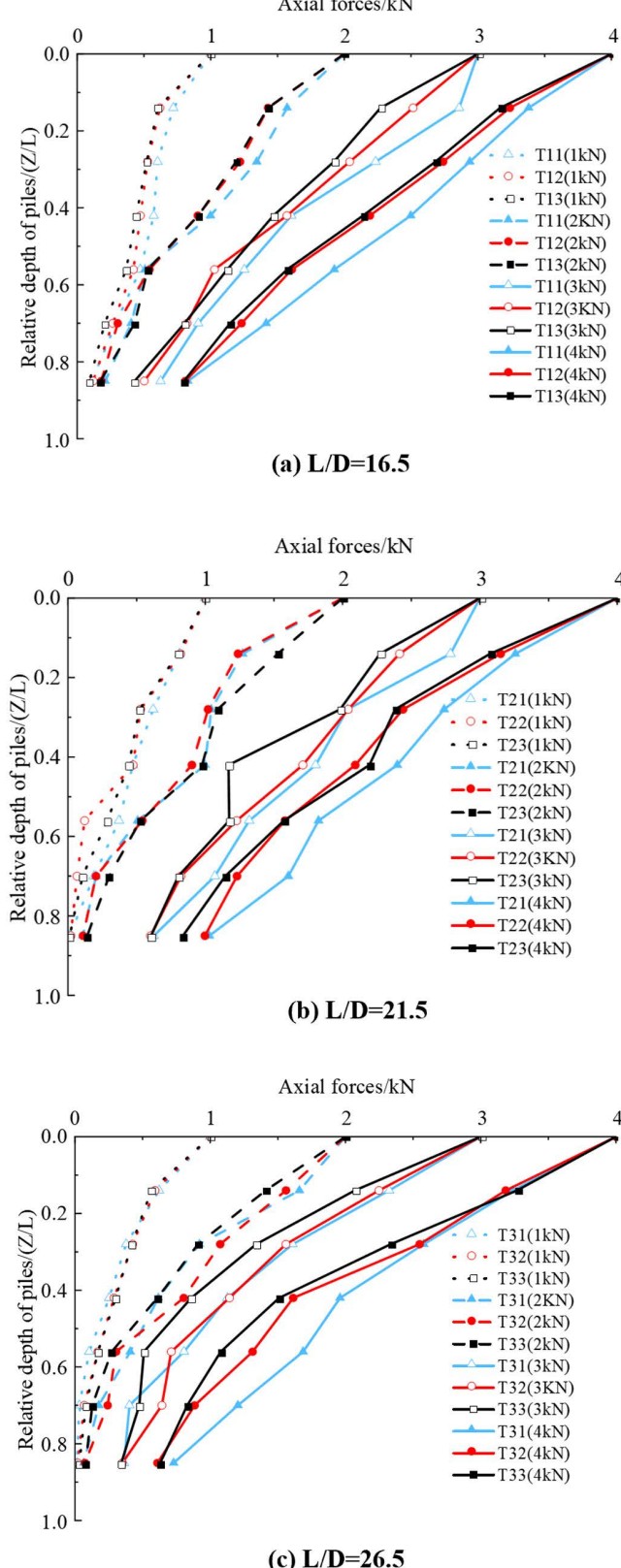

**Fig 8. Variation curve of pile axial force along depth.**

Under the same inclination angle and mechanical load, the rate of change of axial force decreases as the L/D ratio increases and increases as the L/D ratio decreases. For instance, under a 3 kN load, after the pile top settlement stabilizes, the axial force for different groups is as follows: T12: 2.515 kN at Z/L = 0.14 and 0.502 kN at Z/L = 0.85, a reduction of 2.013 kN. T22: 2.415 kN at Z/L = 0.14 and 0.602 kN at Z/L = 0.85, a decrease of 1.813 kN. T32: 2.248 kN at Z/L = 0.14 and 0.351 kN at Z/L = 0.85, a reduction of 1.897 kN. These results show that, as the L/D ratio increases, the axial force reduction becomes smaller. For example, T12 (with a smaller L/D ratio) experiences a more significant reduction in axial force (2.013 kN), while T32 (with a larger L/D ratio) has a more minor reduction (1.897 kN).

## 3.4 Bending moment

The bending moment of the pile can be calculated by formula (2) [41].

$$M_i = \frac{E_p I \Delta \varepsilon_i}{d} \tag{2}$$

where $E_p I$ and $d$ are the bending stiffness of the pile and the diameter of the model pile, respectively, and $\Delta \varepsilon_i$ is the difference between the bending and tensile strains of section I of the pile.

Fig 9 shows that under the same test conditions, the pile bending moment increases to a maximum value with increasing relative depth, then gradually decreases, tending toward zero after a certain depth. Under different working conditions, the bending moment reaches its maximum value of 10.72 N·m at Z/L = 0.24 and decreases near the pile end, approaching zero. When the pile length-to-diameter ratio and inclination angle are more significant, the pile may experience a negative bending moment. For example, when L/D = 26.5 and θ = 20°, the pile exhibits a negative bending moment of -0.13 N·m at Z/L = 0.56. This indicates that both a larger L/D ratio and a greater inclination angle can lead to negative bending moments.

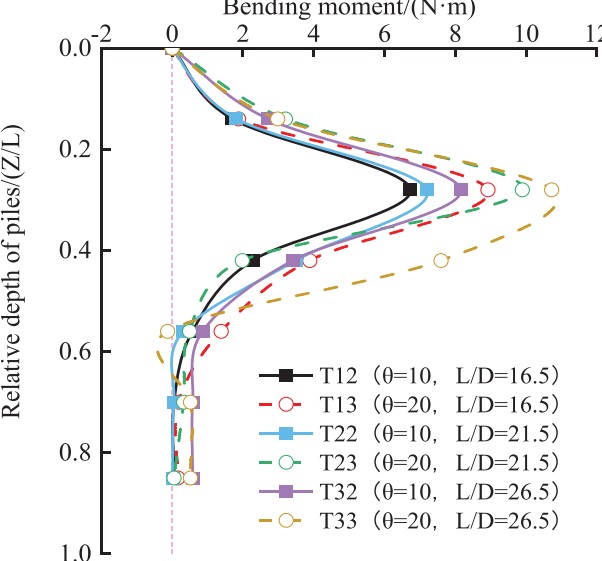

**Fig 9. Change curve of pile bending moment.**

At shallower depths, the larger the inclination angle of the pile, the more pronounced the change in the pile's bending moment when the L/D ratio is the same. For instance, at Z/L = 0.28, the maximum bending moments are: T33: 10.72 N·m, T32: 9.90 N·m (0.82 N·m larger than T32) Similarly: T23: 8.16 N·m, T22: 7.20 N·m (0.96 N·m larger than T22). And for: T13: 8.92 N·m, T12: 6.73 N·m (2.19 N·m larger than T12). This is because a larger inclination angle increases the horizontal component of the force near the top of the pile for a given vertical load. The bending moment is primarily caused by the vertical load acting perpendicular to the pile's longitudinal axis. As the horizontal force increases, the pile tends to produce greater horizontal displacement, leading to overall bending deformation.

For the same inclination angle of the inclined pile, the change of the bending moment of the pile at the same cross-section position is more evident as the length-to-diameter ratio of the pile is larger. For example, at Z/L = 0.28, the bending moment of the pile of group T23 and T33 increased by 10.85% and 20.16%, respectively, compared with group T13.

### 3.5 Shear forces

The shear force of the pile can be calculated using formula (3) [41].

$$F_i = \frac{dM_i}{dx} \tag{3}$$

As shown in Fig 10, the maximum shear force occurs near the top of the pile under different working conditions. This is due to the maximum bending moment generated by the horizontal component of the vertical load at the load application point. This behavior aligns with the findings of Cao [41], who observed that the shear force reaches its maximum at the load application point. Below the soil interface, the shear force initially increases, then decreases and converges to zero.

Under the same test conditions, a more considerable pile inclination results in a higher shear force and a more pronounced rate of change. For the same inclination, a larger L/D ratio

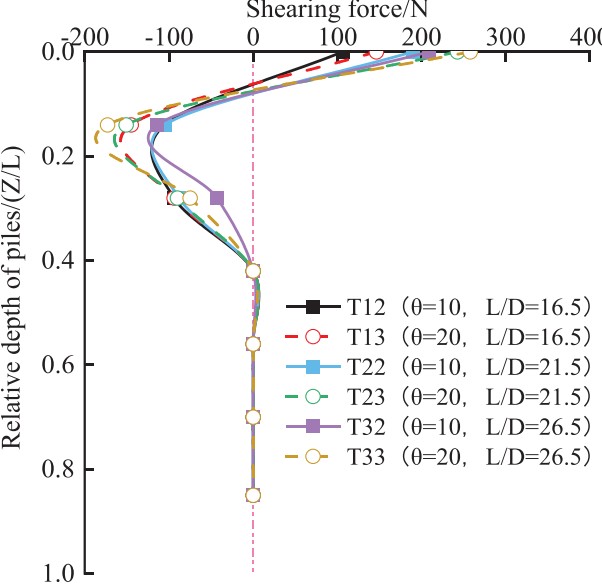

**Fig 10. Variation curve of pile shear force with depth.**

leads to a more significant change in the bending moment. For example, when L/D = 26.5, the maximum shear force near the top of the pile is 258 N for the T33 group and 208 N for the T23 group. Negative shear is observed between Z/L = 0.14 and 0.42, and shear force becomes zero when Z/L > 0.42. This may be due to the high compaction of the red clay soil, which inhibits the effective transfer of the horizontal component force from the top of the pile to the deeper layers.

Under the same conditions, a larger L/D ratio results in a more excellent maximum pile shear. For example, the maximum shear for the T13 pile is 146 N, while the maximum shear for the T33 pile is 1.77 times that of T13. This is consistent with Cao's conclusion [42].

### 3.6 Lateral friction resistance

The lateral friction resistance of the pile can be calculated according to formula (4) [41].

$$f = \left(Q_i - Q_{i-1}\right) / u l_i \tag{4}$$

where $\mu$ is the perimeter of the pile; $l_i$ is the length of the pile section between section $i\text{-}1$ and section $i$ of the pile; $Q_{i\text{-}1}$, $Q_i$ section $i\text{-}1$ and axial force at section $i, i = 1,2,3,4,\ldots$ indicates from the top of the pile to the bottom of the pile.

As the top of the pile is subjected only to vertical mechanical loads, the lateral resistances are all positive. As shown in Fig 11, the lateral friction resistance generally decreases with the increase in adequate depth. It reaches its maximum near the top of the pile, while near the bottom, it is generally smaller.

Comparing Fig 11(a)–(c), it can be seen that the average pile-side friction resistance at different depths increases with the increase of pile-top load, regardless of whether the pile is inclined or vertical. When the vertical load at the top of the pile is small, the average lateral friction resistance is mainly concentrated in the upper section of the pile, with minor resistance in the lower section. In the upper section, inclined piles exhibit more excellent average side friction resistance than vertical piles. In contrast, in the lower section, the difference in resistance between inclined and vertical piles is minimal. This suggests that the friction in the upper section of the inclined pile is more fully utilized, while the friction in the lower section of the vertical pile is more fully utilized. This difference explains why vertical displacement occurs only in vertical piles under vertical load, whereas both horizontal and vertical displacements occur in inclined piles.

## 4 Conclusion

(1) Vertical Bearing Performance: When the vertical load at the top of the pile and the length-to-diameter (L/D) ratio are fixed, vertical piles exhibit the best vertical bearing performance, resulting in less settlement compared to inclined piles with angles of 10° and 20°. However, when both the vertical load and inclination angle are fixed, and with an L/D ratio of 26.5, inclined piles perform better in terms of vertical bearing capacity, generating less settlement than inclined piles with L/D ratios of 16.5 and 21.5.

(2) Bending Moment Distribution: The bending moment of the pile reaches its maximum value at Z/L = 0.24 for all working conditions. The larger the inclination angle of the pile at shallower depths, the more pronounced the change in the bending moment distribution for the same L/D ratio. As the depth increases, the bending moment gradually decreases and tends towards zero.

(3) Shear Force Distribution: The maximum shear force in an inclined pile occurs near the top of the pile. An extreme shear force is observed at a certain depth below the top,

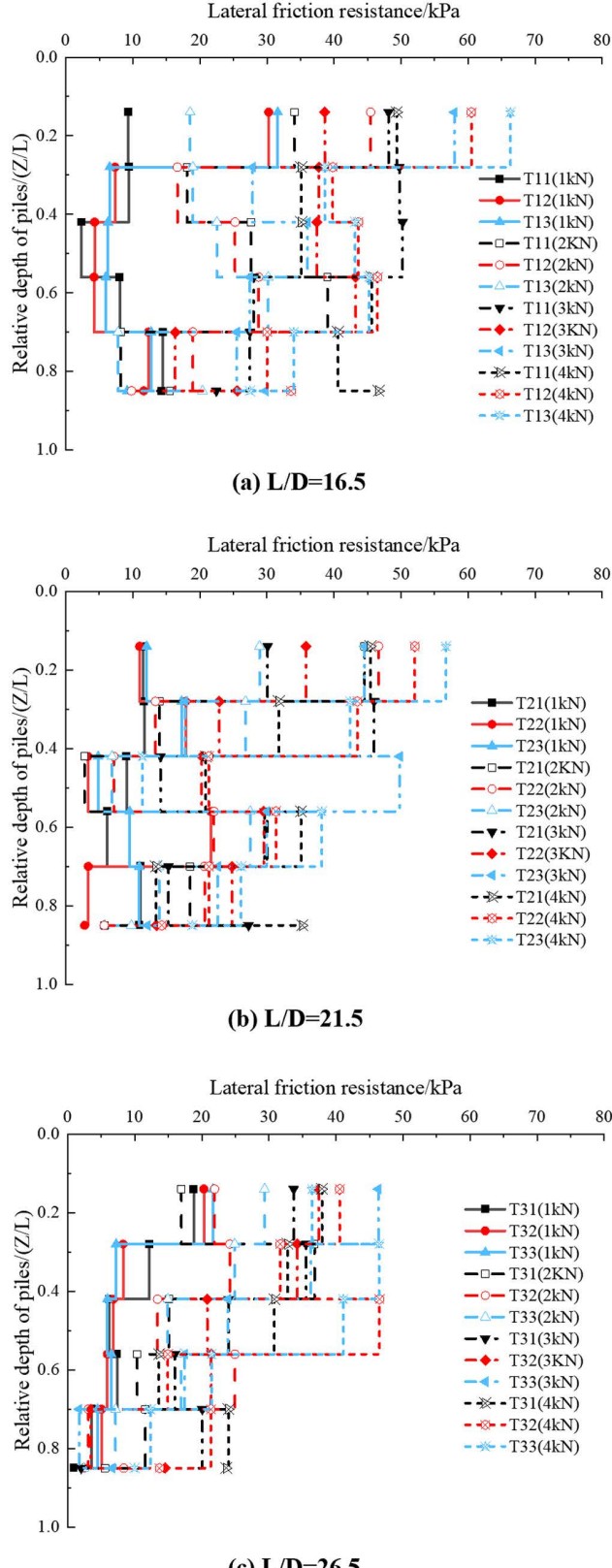

(a) L/D=16.5

(b) L/D=21.5

(c) L/D=26.5

Fig 11. Variation curve of lateral friction resistance of pile along the depth.

roughly at Z/L = 0.14 in this study. The shear force increases with both the L/D ratio and pile inclination, indicating that a larger L/D ratio and inclination result in higher shear forces at the same depth.

(4) Frictional Resistance: Regardless of the vertical load, pile inclination, and L/D ratio, the maximum lateral frictional resistance in inclined piles occurs in the section near the top of the pile, typically at a relative depth of Z/L = 0.14. In the upper section of the pile, the average side friction resistance is more significant for inclined piles than for vertical piles. However, in the lower section, the difference in average lateral friction resistance between inclined and vertical piles is generally tiny.

It should be noted that in this paper, compressive tests were conducted only on inclined piles under free conditions at the top of the pile in red clay with certain water content and heaviness, and the effect of groundwater was not considered. More tests are needed to investigate the compressive properties of inclined piles in other states of cohesive soils as well as in cohesionless soils with different displacement boundary conditions at the top of the pile.

## Supporting information

**S1 Data. Minimum data set**
(DOCX)

## Author contributions

**Funding acquisition:** Fei Gan.

**Writing – original draft:** Shilang Guo.

**Writing – review & editing:** Shilang Guo, Fei Gan, Hong Wang, Jing Bi, Biao Liu, Yuanyin Zhang.

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
