## [Decision Letter · Decision Letter 0]

6 Nov 2024

PONE-D-24-43810Research on vertical bearing performance of inclined high-pressure rotary spray pilePLOS ONE

Dear Dr. Gan,

Thank you for submitting your manuscript to PLOS ONE. After careful consideration, we feel that it has merit but does not fully meet PLOS ONE’s publication criteria as it currently stands. Therefore, we invite you to submit a revised version of the manuscript that addresses the points raised during the review process.

**It is recommended that the manuscript should be revised properly and deal the all comments raised in the review comments.    **

We look forward to receiving your revised manuscript.

Kind regards,

Ghulam Yaseen, Ph.D.

Academic Editor

PLOS ONE

**Journal Requirements:**

Reviewers' comments:

Reviewer's Responses to Questions

**Comments to the Author**

1. Is the manuscript technically sound, and do the data support the conclusions?

Reviewer #1: No

Reviewer #2: No

2. Has the statistical analysis been performed appropriately and rigorously? 

Reviewer #1: No

Reviewer #2: N/A

3. Have the authors made all data underlying the findings in their manuscript fully available?

Reviewer #1: No

Reviewer #2: Yes

4. Is the manuscript presented in an intelligible fashion and written in standard English?

Reviewer #1: No

Reviewer #2: Yes

5. Review Comments to the Author

**Reviewer #1:**  Sorry, this work is REJECTED. Its structure doesn’t follow the IMRAD. Basically, it is a simple report of some lab test to employer for a small scale case study (However the standard used is opaque).

Trivial unbalanced. Only Introduction has references and the rest of work 21 pages without any documentation???????

Overall:

1. Poor English

2. Long Abstract, unreflective and general keywords

3. Superficial technical review: Uncharacterized research gaps/not critically analyzed relevant updated works/not outlined the main innovation level in the science border in comparison with other studies.

4. Which type of foundation???

5. Doesn’t have any convincing and documented Discussion in terms of evidential analysis/solid comparison with other scholars’ approaches/uncertainty quantification/technical limitations/pitfalls and practical difficulties/the impact of the used data and involved noises on the results/ …

6. Doesn’t consider the spatial soil type distribution mapping and corresponding effects on the results (for example, https://link.springer.com/article/10.1007/s10064-018-1400-9...)

7. The effect of ground water is neglected.

8. The effect of clay sensitivity on engineering results is not evaluated (for example https://www.sciencedirect.com/science/article/abs/pii/S0013795215000411, https://link.springer.com/article/10.1007/s10706-016-9976-y …)

9. Ill-formatted reference list in terms of format, identifiers, DOI…

10. Unjustified conclusion

**Reviewer #2: ** 1- Rewrite the abstract so that it is limited to presenting the problem of the research and the results reached by the researcher.

2.1

a- What are the criteria for selecting dimensions of soil tank؟

b- Why didn't the researcher use natural soil in the physics model?

2.3

a- The model piles were made by cast-in-place slurry reinforcement in hydraulic soil ??

Table 2

Srandard for each test??

Fig.4 where is the clay? Note: the soil sample is sand according to grain size distribution curve

3.2 3.2 Vertical bearing capacity to 3.2 bearing capacity

Figures 7, change the Y axis title to Settlement

3.3

Eq. 1 (Ref.?)

4 Conclusion

Rewrite the conclusions to include the most important numerical findings of the study

6. PLOS authors have the option to publish the peer review history of their article (what does this mean? ). If published, this will include your full peer review and any attached files.

**Do you want your identity to be public for this peer review?** For information about this choice, including consent withdrawal, please see our Privacy Policy .

Reviewer #1: No

Reviewer #2: No

---

## [Author Response · Author response to Decision Letter 1]

30 Nov 2024

Response to Academic Editorial Editor:

Thank you for the editor's suggestion. We have made revisions to the paper.

Responds to the reviewer#1’s comments:

Thank you for the reviewer's comments. We have made revisions to the paper.

Responds to the reviewer#2’s comments:

We are very grateful for the reviewer's reading and suggestions, and we have made revisions to the paper.

---

## [Decision Letter · Decision Letter 1]

30 Dec 2024

PONE-D-24-43810R1Research on vertical bearing performance of inclined high-pressure rotary spray pilePLOS ONE

Dear Dr. Gan,

Thank you for submitting your manuscript to PLOS ONE. After careful consideration, we feel that it has merit but does not fully meet PLOS ONE’s publication criteria as it currently stands. Therefore, we invite you to submit a revised version of the manuscript that addresses the points raised during the review process.

We look forward to receiving your revised manuscript.

Kind regards,

Ghulam Yaseen, Ph.D.

Academic Editor

PLOS ONE

Reviewers' comments:

Reviewer's Responses to Questions

**Comments to the Author**

1. If the authors have adequately addressed your comments raised in a previous round of review and you feel that this manuscript is now acceptable for publication, you may indicate that here to bypass the “Comments to the Author” section, enter your conflict of interest statement in the “Confidential to Editor” section, and submit your "Accept" recommendation.

Reviewer #1: (No Response)

Reviewer #2: (No Response)

2. Is the manuscript technically sound, and do the data support the conclusions?

Reviewer #1: Partly

Reviewer #2: Yes

3. Has the statistical analysis been performed appropriately and rigorously? 

Reviewer #1: No

Reviewer #2: Yes

4. Have the authors made all data underlying the findings in their manuscript fully available?

Reviewer #1: No

Reviewer #2: Yes

5. Is the manuscript presented in an intelligible fashion and written in standard English?

Reviewer #1: No

Reviewer #2: Yes

6. Review Comments to the Author

Reviewer #1: Thanks for the responses. However, ‘Conclusion and Discussion’ are two different stories . You have discussed your results so; this part should be ‘conclusion’ or ‘concluding remarks’.

Some inconsistencies within the reference list MUST be treated. 26 and 29 are the same and one of them MUST be removed or replaced. Double check the rest of references.

Change the title and remove ‘search’.

Please doublecheck the English to polish some minor typos

Reviewer #2: Dear Auther(s)

Fig. 4 (According to the grain size distribution, the soil is sand, not clay).Please correct the curve or indicate the limits of both sand and clay size, citing the approved standard.

7. PLOS authors have the option to publish the peer review history of their article (what does this mean? ). If published, this will include your full peer review and any attached files.

**Do you want your identity to be public for this peer review?** For information about this choice, including consent withdrawal, please see our Privacy Policy .

Reviewer #1: No

Reviewer #2: **Yes: ** Maki Jafar Al-Waily

---

## [Author Response · Author response to Decision Letter 2]

3 Jan 2025

Reviewer #1:

1. Thanks for the responses. However, ‘Conclusion and Discussion’ are two different stories. You have discussed your results so; this part should be ‘conclusion’ or ‘concluding remarks’.

Response: Thank you very much. We have made the necessary modifications based on your feedback

（See line 345 on page 17 in the article）

2. Some inconsistencies within the reference list MUST be treated. 26 and 29 are the same and one of them MUST be removed or replaced. Double check the rest of references.

Response: Thank you very much for the reviewer's reminder. We have replaced Article 29. And checked the remaining references.

[29]. Shahri A A, Moud F M (2021) Landslide susceptibility mapping using hybridized block modular intelligence model. BULLETIN OF ENGINEERING GEOLOGY AND THE ENVIRONMENT. 80(1) 267-284. https://doi.org/10.1007/s10064-020-01922-8.

（See lines 458-460 on page 21 in the article）

3. Change the title and remove ‘search’.

Response: Thank you very much. We have revised the title based on your feedback.

‘Vertical bearing performance of inclined high-pressure rotary spray pile’

（See line 1 on page 1 in the article）

4. Please doublecheck the English to polish some minor typos.

Response: Thank you very much for your suggestion. We have made every effort to improve the manuscript. We have carefully checked the English and eliminated some minor spelling errors. We appreciate the reviewer's diligent work and hope the corrections will be approved.

Reviewer # 2:

1．Dear Auther(s)

Fig. 4 (According to the grain size distribution, the soil is sand, not clay).Please correct the curve or indicate the limits of both sand and clay size, citing the approved standard.

Response: Dear reviewer, thank you very much for your reminder. We have remeasured and recorded the particle size distribution data of the red clay, and plotted the grading curve of the red clay. We hope for your approval.

（See line 121 on page 6 in the article）

---

## [Decision Letter · Decision Letter 2]

29 Jan 2025

PONE-D-24-43810R2Vertical bearing performance of inclined high-pressure rotary spray pilePLOS ONE

Dear Dr. Gan,

Thank you for submitting your manuscript to PLOS ONE. After careful consideration, we feel that it has merit but does not fully meet PLOS ONE’s publication criteria as it currently stands. Therefore, we invite you to submit a revised version of the manuscript that addresses the points raised during the review process.

We look forward to receiving your revised manuscript.

Kind regards,

Ghulam Yaseen, Ph.D.

Academic Editor

PLOS ONE

Journal Requirements:

Reviewers' comments:

Reviewer's Responses to Questions

**Comments to the Author**

1. If the authors have adequately addressed your comments raised in a previous round of review and you feel that this manuscript is now acceptable for publication, you may indicate that here to bypass the “Comments to the Author” section, enter your conflict of interest statement in the “Confidential to Editor” section, and submit your "Accept" recommendation.

Reviewer #1: All comments have been addressed

2. Is the manuscript technically sound, and do the data support the conclusions?

Reviewer #1: Partly

Reviewer #2: Yes

3. Has the statistical analysis been performed appropriately and rigorously? 

Reviewer #1: Yes

Reviewer #2: Yes

4. Have the authors made all data underlying the findings in their manuscript fully available?

Reviewer #1: No

Reviewer #2: Yes

5. Is the manuscript presented in an intelligible fashion and written in standard English?

Reviewer #1: No

Reviewer #2: Yes

6. Review Comments to the Author

Reviewer #1: Thanks for your cooperation and responding. Before publication, make a review to remove some minor typos.

Good luck

Reviewer #2: Dear Autheor(s)

In figure 4, the soil is gravelly sand, not clay soil, according to Unified Soil Classification System (ASTM D2487-17e1)

Gravel (>4.75 mm) = 100-64 = 36%

Sand = 64%

Clay = 0 %

Please redraw the figure so that the passage through the No. 200 sieve is more than 50% according to the unified classification system, or set the grain sizes of gravel, sand and clay according to Chinese specifications

7. PLOS authors have the option to publish the peer review history of their article (what does this mean? ). If published, this will include your full peer review and any attached files.

**Do you want your identity to be public for this peer review?** For information about this choice, including consent withdrawal, please see our Privacy Policy .

Reviewer #1: No

Reviewer #2: **Yes: ** Maki J. Mohammed Al-Waily

---

## [Author Response · Author response to Decision Letter 3]

7 Feb 2025

Responding to the editor's comments:

Response: Thanks for the editor's reminder. We have checked the reference list to ensure its completeness and accuracy.

Responds to the reviewer’s comments:

Reviewer #1:

1: Thanks for your cooperation and responding. Before publication, make a review to remove some minor typos.

Good luck

Response: Thank you very much for the patient response from the reviewer and also for your blessings. We have carefully checked the grammar in the article and removed some minor spelling errors. The modified parts have been indicated in red font in the article(Revised Manuscript with Track Changes). We hope the reviewer approves.

Reviewer # 2:

1. Dear Author(s)

In figure 4, the soil is gravelly sand, not clay soil, according to Unified Soil Classification System (ASTM D2487-17e1)

Gravel (>4.75 mm) = 100-64 = 36%

Sand = 64%

Clay = 0 %

Please redraw the figure so that the passage through the No. 200 sieve is more than 50% according to the unified classification system, or set the grain sizes of gravel, sand and clay according to Chinese specifications

Response: Thank you very much for the reviewer's serious and patient response. We have redrawn the chart according to the reviewer's request, and we sincerely hope to receive your approval.

（See line 120 on page 6 in the article）

---

## [Editor Report · Decision Letter 3]

10 Feb 2025

Vertical bearing performance of inclined high-pressure rotary spray pile

PONE-D-24-43810R3

Dear Dr. Gan,

We’re pleased to inform you that your manuscript has been judged scientifically suitable for publication and will be formally accepted for publication once it meets all outstanding technical requirements.

Kind regards,

Ghulam Yaseen, Ph.D.

Academic Editor

PLOS ONE
---

## [Editor Report · Acceptance letter]

PONE-D-24-43810R3

PLOS ONE

Dear Dr. Gan,

I'm pleased to inform you that your manuscript has been deemed suitable for publication in PLOS ONE. Congratulations! Your manuscript is now being handed over to our production team.

Kind regards,

on behalf of

Professor Ghulam Yaseen

Academic Editor

PLOS ONE